# Modification of Inconel 718 Properties by In Situ Y Addition in Selective Laser Melting

**DOI:** 10.3390/ma15186219

**Published:** 2022-09-07

**Authors:** Evgenii Borisov, Anatoly Popovich, Vadim Sufiiarov

**Affiliations:** Institute of Machinery, Materials, and Transport, Peter the Great St.Petersburg Polytechnic University, Polytechnicheskaya, 29, 195221 Saint Petersburg, Russia

**Keywords:** selective laser melting, additive manufacturing, Inconel, in situ, superalloy, rare earth metal, yttrium

## Abstract

The paper presents the results of a study of the possibility of Inconel 718 alloy properties modifying by adding Yttrium in situ in the process of selective laser melting. The single and double laser processing of each layer was used. Yttrium was introduced into the alloy in an amount of 0.1, 0.2, 0.5, 1, and 2 mass %. Studies of the structure of the material showed that undissolved Yttrium particles remain in the material. With an increase in the proportion of yttrium in the alloy, the hardness increases. Tensile test showed that with an increase in the content of yttrium from 0 to 0.1%, the yield strength and tensile strength decrease, but the relative elongation increases. With a further increase in the yttrium content, there is a sharp decrease in the relative elongation and an increase in the yield strength, which is caused by the existence of a large number of undissolved yttrium particles in the sample.

## 1. Introduction

Inconel 718 is one of the high temperature superalloys suitable for many applications. This alloy has good corrosion resistance, high strength and weldability. Alloy Inconel 718 is designed to operate at temperatures up to 700 °C [1,2,3]. Nickel superalloys are widely used in the manufacture of parts for gas turbines in the aerospace, shipbuilding, automotive and energy industries. They have high strength at elevated temperatures, high fatigue properties, wear resistance, and corrosion resistance, which makes it possible to use products made from these materials in aggressive environments [2,3].

The addition of rare earth elements to alloys, especially heat-resistant nickel alloys, has long been used in their production. There are works that study the addition of yttrium to alloys in order to improve their characteristics [1,2,3,4,5,6,7]. There is evidence of an increase in strength during tensile testing, the purification of alloys from sulfur and oxygen, and the effect on the morphology of grains and carbides as well as more prominent precipitation for γ″ [2,3,4,8,9]. However, studies show that up to a certain concentration, the addition of yttrium has a positive effect on the structure and properties, but when this concentration is exceeded, the properties of the material decrease. This is caused by the formation of yttrium-rich phases (Ni_17_Y_2_ intermetallic compounds, yttrium oxides and nitrides) [2,5,10]. Yttrium binds oxygen and sulfur contained in the melt well; however, during casting, this can lead to a reaction of the alloy with the material of the mold and the formation of casting defects. Due to their relatively high affinity for oxygen, nitrogen, and sulfur, yttrium compounds have a lower standard energy of formation. In addition, due to the higher melting temperature (for Y_2_O_3_, 2400 °C), such compounds can be present in the superalloy melt and precipitate along the boundaries, increasing their strength and reducing the speed of grain boundaries movement. In superalloys, Y is often present in the carbides, increasing their lattice parameters and increasing the misfit between the carbide and the matrix. This leads to a decrease in the growth rate of carbides, their size and a change in their morphology to block, increasing the resistance to cracking. An increased yttrium content can lead to the formation of Al_2_Ni_6_Y_3_ along grain boundaries, which can be a source of crack initiation [4,5,6,7].

The strengthening of superalloys by dispersed particles introduced into the alloy is also one of the important topics [11,12]. Alloys, dispersion strengthened using oxides, are often used in high-temperature applications, such as gas turbine engines, chemical reactors, etc. Due to the chemical resistance and high melting point of such oxides, they remain stable, even at high temperatures [13].

Additive technologies, in particular, technologies of selective laser melting (SLM), make it possible to manufacture metal products of complex geometric shapes, which are often impossible to obtain by traditional methods. SLM is a multifactorial technology in which the structure and properties of the resulting products depend on the initial parameters [14]. This technology makes it possible to obtain products of a complex shape without the need to manufacture complex tooling, casting molds, etc. After manufacturing, a workpiece is obtained that is closest to the shape of the final part.

An important advantage is the use of powder as an initial material. This opens up the possibility of using powder mixtures and composition control, as well as the synthesis of alloys in the SLM process [15]. Taking into account the effects that occur in superalloys with the addition of pure yttrium, as well as its compounds, in this work, it was decided to use yttrium metal in the form of a powder to be added to a nickel superalloy. Thus, yttrium in the superalloy can play a role both in the metallic form and in the form of oxides and other compounds, both present on the surface of yttrium particles and formed during SLM.

Thus, the purpose of this work was to study the process of selective laser melting of a mixture of Inconel 718 powders and metallic Y and the properties of the formed material.

## 2. Materials and Methods

The initial material was Inconel 718 heat-resistant alloy powder produced by the technology of melt gas atomization by LPW Technology Ltd (Runcorn, United Kingdom). Yttrium powder (OChV, Moscow, Russia) was used as a rare earth metal powder. The morphology of the powders’ particles is shown in Figure 1.

The morphology of the powder was studied using a scanning electron microscope (SEM, Tescan Mira 3).

The powders were mixed with a gravity mixer. Each composition was mixed for 12 h.

Samples were fabricated using the selective laser melting technology on an SLM 280HL machine (SLM Solutions GmbH, Luebeck, Germany) in a nitrogen atmosphere. The values of the technological parameters for the manufacture of samples were power, 250 Watts; scanning speed, 700 mm/s; the distance between the individual laser passes, 0.12 mm; the layer thickness, 0.05 mm; and the diameter of the laser beam, 0.08 mm. This parameter set provides sufficient density of the material during manufacture, as shown by earlier studies [16]. The study was carried out both for a single laser treatment and for a double one, according to the same parameter set. Secondary laser processing was carried out without intermediate recoating of the powder material. The density of the samples was studied using the method of hydrostatic weighing (Archimedes’ principle).

Tensile samples were made in a horizontal orientation (perpendicular to the growth direction). Samples made in this way usually have higher strength values but lower relative elongation [17,18].

The hardness of the samples was measured using the Zwick/Roell ZHU 250 tester (Zwick GmbH, Ulm, Germany) with a Vickers indenter.

Tensile testing of the specimens was carried out on a Zwick/Roell Z100 testing machine.

X-ray diffractometer Bruker D8 Advance was used for X-ray analysis using Cu Kα (1.5406 Å) radiation.

The powder size distribution of the obtained powder was investigated by the laser diffraction method (ISO 13320:2020), using a laser diffractometer (Fritsch Analysette 22, FRITSCH GmbH, Idar-Oberstein, Germany).

The study of the microstructure of samples was carried out using an optical microscope Leica Dmi8C.

Heat treatment of the samples was carried out using a vacuum furnace (Carbolite Gero GmbH & Co. KG, Neuhausen, Germany) as follows:-Annealing at 1065 °C for 1 h and subsequent gas cooling;-Two-stage aging: heating to 760 °C with holding for 10 h, then cooling to 650 °C within 2 h and then holding at 650 °C for 8 h followed by gas cooling.

## 3. Results

### 3.1. Powder Materials Characteristics

As can be seen in Figure 1 and Figure 2, the powder has a particle shape close to spherical. There are oval and elongated particles. In addition, there are agglomerates of powder particles. At the same time, the sizes and shape of the particles of Inconel 718 and yttrium powders are similar. The d_50_ size for the Inconel 718 powder is 47 µm, and for the yttrium powder, 43 µm.

Traditionally, additions of rare earth metals to the alloy are fractions of a percent [1,2,3]. However, due to the fact that the particle size of both powders is quite large, in this case, it can be quite difficult to add a small fraction of the powder and ensure its uniform distribution, when adding even 0.5 wt. % of yttrium powder, resulting in a particle ratio of about 1 Y particle per 100 superalloy particles. With an average particle size of the powder material of about 40–50 µm, it turns out that one particle of yttrium will not be contained in every volume of the material processed by the laser beam, and the uniformity of the distribution of yttrium in the material will strongly depend on the uniformity of the mixing of the initial powders. Therefore, in addition to a single laser treatment of each layer, a double treatment was also applied in the work. Such treatment should lead to a more uniform distribution of yttrium in the alloy and a more complete dissolution of yttrium particles.

As values for the proportion of yttrium added to the alloy, 0.1%, 0.2%, 0.5%, 1% and 2% by mass were chosen. For each composition, a mixture was prepared, and samples were made. The technological characteristics of the resulting mixture (flow rate and bulk density) remained practically unchanged due to the fact that the size and shape of the particles of the powders are almost the same, and the difference in material density does not affect due to the small proportion of yttrium particles in the total mixture.

### 3.2. Selective Laser Melting

Next, samples were grown from a mixture of powders. For this, samples were made with dimensions of 10x10x10 mm for each parameter set. Additionally, samples were made with double laser processing without deposition of a new layer. After that, the density of the obtained samples was examined. The results of the study of the samples density are presented in Table 1.

Sample studies have shown that the density of samples with the addition of yttrium is slightly reduced. This may be due to the fact that the density of yttrium is less than that of Inconel 718. In addition, a decrease in the density of the material can be caused by the formation of defects in the form of pores. The image and distribution of pores in the microstructure of the samples is shown in Figure 2. With an increase in the proportion of yttrium in the alloy, an increase in the number and size of pores is noticeable. With the addition of 0.5% Y, the fraction of pores on the microsection is 0.5–0.7, and with the addition of 2% Y, it is 0.8–1%.

A possible reason for the formation of such defects is the insufficient wettability of the surface of yttrium particles coated with oxide by the Inconel 718 melt [19]. In this case, as can be seen from Table 1, the density of the material in some samples increases with the use of double laser processing. This is due to the repeated melting of the material and the possible release of pores from the melt [20]. Based on the data obtained, further studies were carried out only for samples using double laser processing.

### 3.3. Scanning Electron Microscopy with Energy Dispersive Spectroscopy

Studies using SEM and EDS showed that in the structure of the samples, regardless of the percentage of yttrium, there are undissolved yttrium particles. With an increase in the proportion of yttrium in the initial powder, the proportion of such particles in the samples increases. This indicates that the use of double scanning only leads to a decrease in the size of individual undissolved particles, but they do not completely disappear. After heat treatment, the remaining particles of undissolved yttrium powder were still found on the sections. An example of EDS images for samples with 1% Y before and after HT is shown in Figure 3.

An XRD study was carried out to study the phase composition of the material. The XRD curves are shown in Figure 4. The curves are shown only for samples made from a mixture of powders with 1 and 2% Y, since the remaining compositions, due to the low content of yttrium, do not show any distinguishable peaks, except for the phases inherent in the alloy Inconel 718.

As can be seen from the curves, there are no peaks corresponding to pure yttrium, as well as to yttrium oxide. A sample containing 2% yttrium has a single small peak, which may correspond to yttrium nitride (YN) [21]. However, due to its small size and the absence of reflections at other characteristic angles, this cannot be confirmed reliably for this compound.

### 3.4. Mechanical Properties

Data on the study of the hardness of samples after heat treatment are presented in Table 2. With an increase of the yttrium content in the alloy, the hardness values increase. This may be caused by an increased content of γ″ in a material with a higher content of yttrium. It is known that when yttrium is added, the γ″ phase initiation and peak temperatures decrease, which leads to higher precipitation volume fraction [9].

At the next stage of the study, the mechanical properties of the samples were tested in tension. The results of studying the tensile properties of samples are shown in Table 3. With an increase in the yttrium content in the alloy to 0.1–0.2%, first, a slight decrease in the tensile strength and yield occurs, while at the same time, the relative elongation slightly increases, which is consistent with the previously discovered dependence in [1]. However, with a further increase in the yttrium content, an increase in the yield strength of the samples occurs with a simultaneous significant decrease in the relative elongation. An increase in the yield strength may be due to the precipitation of γ″. The Inconel 718 + 2% Y sample showed the best yield strength results. In this case, for all samples, the tensile strength exceeds 1300 MPa.

When more than 0.1% yttrium is added, the value of relative elongation for the studied compositions drops sharply, down to 2%. This may be a consequence of the presence of pores in the structure of the samples, as well as undissolved particles. To confirm this, a fracture of the sample with the highest yttrium content was studied. A fractography of the Inconel 718 + 2% Y sample is shown in Figure 5.

On the fracture, Yttrium particles (lighter in the BSE image) are clearly visible, not completely dissolved during the selective laser melting process. They could be the centers of crack initiation due to the lower strength properties and cause a decrease in relative elongation at concentrations more than 0.1%Y. The fractography of the rest area revealed an extensive dimple morphology characteristic of a ductile fracture of Inconel 718 superalloy.

Thus, as it shown in Table 2 and Table 3 that it is possible to increase the yield strength and hardness of the alloy with the addition of yttrium. However, in the case of the complete dissolution of the particles, the results may differ. Based on this, further research will be carried out using a finer powder, which will possibly have time to dissolve in the selective laser melting process. However, the large surface area of the small particles and the corresponding higher possible content of yttrium oxide must be taken into account. In addition, it is necessary to optimize the values of technological parameters to reduce porosity, as well as the more complete dissolution of particles. In addition, further research will focus on testing the material at elevated temperatures, including long-term strength and fatigue, to fully understand the effect of adding yttrium on the performance of modified Inconel 718 alloy fabricated by selective laser melting.

## 4. Conclusions

In this work, the influence of the rare earth metal yttrium added to the high-temperature nickel alloy Inconel 718 by in situ synthesis in the process of selective laser melting was considered.
It was found that when using the same parameter sets of selective laser melting leads to increasing porosity in accordance with the yttrium content. Double laser processing allows to reduce the overall porosity and increase the absolute density of the samples.Undissolved yttrium particles were found in all samples. Double laser processing did not lead to the complete disappearance of all undissolved particles. Heat treatment also did not contribute to their full dissolution.Values of the hardness of samples are raised in accordance with the yttrium content, which is due to the increasing fraction of the γ″ phase.With an increase in the content of yttrium from 0 to 0.1%, the yield strength and tensile strength decrease, but the relative elongation increases. With a further increase in the yttrium content, there is a sharp decrease in the relative elongation and an increase in the yield strength.

## Figures and Tables

**Figure 1 materials-15-06219-f001:**
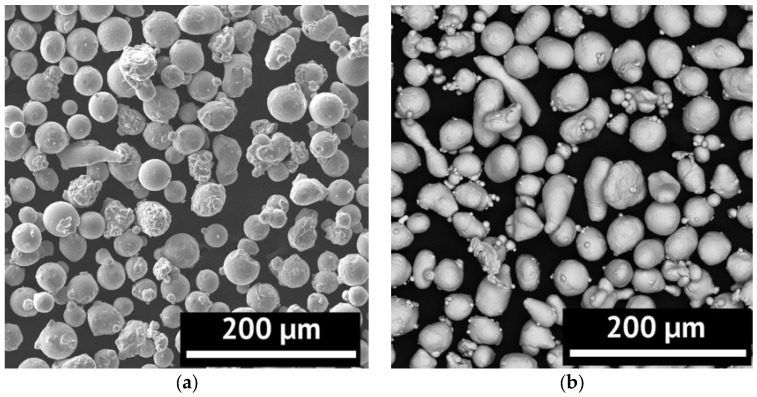
Particle morphology of the initial Inconel 718 (**a**) and yttrium (**b**) powder particles.

**Figure 2 materials-15-06219-f002:**
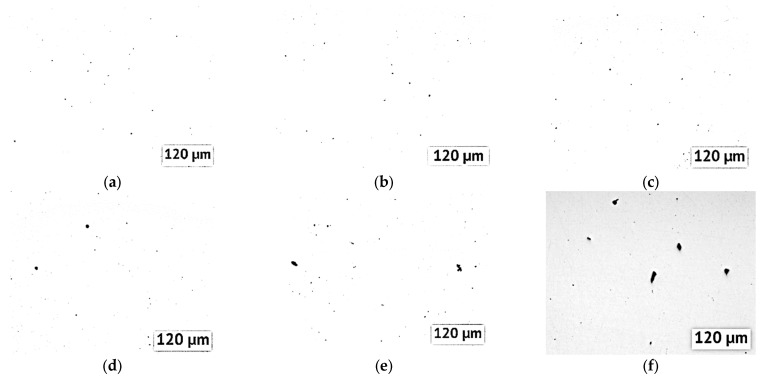
Porosity of the samples without Y (**a**), with 0.1%Y (**b**), with 0.2%Y (**c**), with 0.5%Y (**d**), with 1%Y (**e**), with 2%Y (**f**) after single laser scan. Pores are marked in black.

**Figure 3 materials-15-06219-f003:**
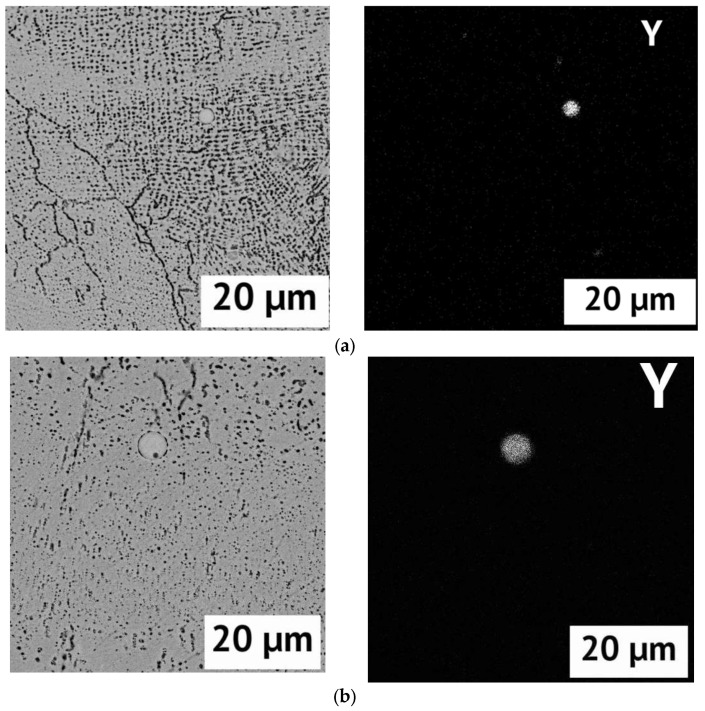
SEM images (left) and corresponding EDS maps of Yttrium distribution (right) of samples with 1% Y (double laser processing): (**a**) after SLM; (**b**) after heat treatment.

**Figure 4 materials-15-06219-f004:**
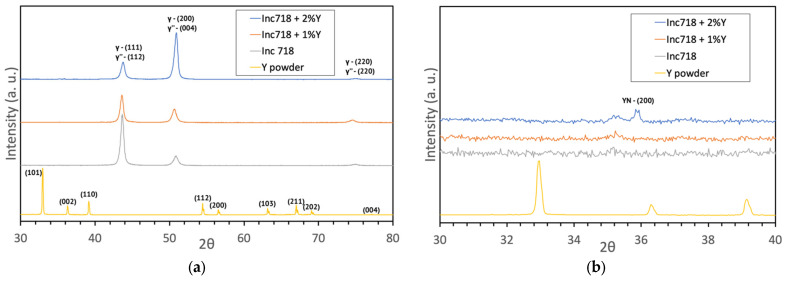
X-ray diffraction patterns of the processed samples (**a**); enlarged part of the graph (**b**).

**Figure 5 materials-15-06219-f005:**
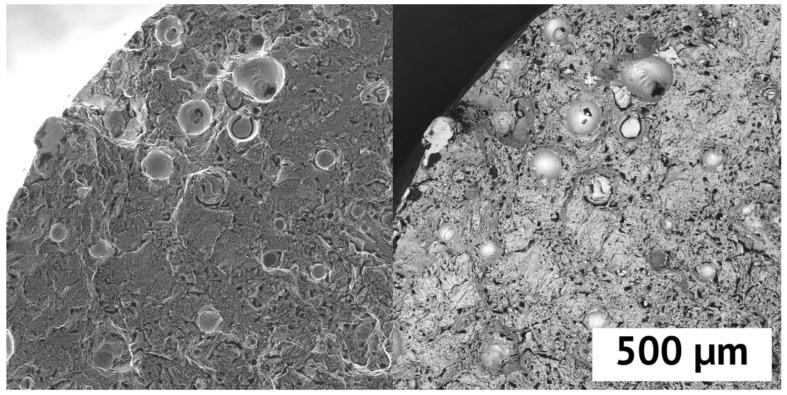
SEM image of the fracture of the sample Inconel 718 + 2% Y. (**Left**)–SE, (**right**)–BSE.

**Table 1 materials-15-06219-t001:** Results of the density measurements of the samples.

Specimen	Density, g/cm^3^	
	1 scan	2 scans
Inconel 718	8.19	8.19
Inconel 718 + 0.1% Y	8.13	8.15
Inconel 718 + 0.2% Y	8.15	8.17
Inconel 718 + 0.5% Y	8.09	8.10
Inconel 718 + 1% Y	8.10	8.12
Inconel 718 + 2% Y	8.10	8.10

**Table 2 materials-15-06219-t002:** Hardness HV_10_ of the samples.

Specimen	Hardness HV_10_
Inconel 718	443
Inconel 718 + 0.1% Y	446
Inconel 718 + 0.2% Y	470
Inconel 718 + 0.5% Y	501
Inconel 718 + 1% Y	511
Inconel 718 + 2% Y	530

**Table 3 materials-15-06219-t003:** Tensile mechanical properties of the samples.

Specimen	Yield Strength, MPa	Ultimate Strength, MPa	Elongation, %
Inconel 718	1180 ± 12	1372 ± 11	13 ± 1
Inconel 718 + 0.1%Y	1170 ± 3	1370 ± 19	15 ± 3
Inconel 718 + 0.2%Y	1150 ± 3	1350 ± 22	11.3 ± 3
Inconel 718 + 0.5%Y	1230 ± 2	1340 ± 7	4.3 ± 1
Inconel 718 + 1%Y	1240 ± 10	1320 ± 17	2 ± 1
Inconel 718 + 2%Y	1260 ± 9	1310 ± 10	2.1 ± 1

## Data Availability

The main data had been provided in the paper. Any other raw/processed data required to reproduce the findings of this study are available from the corresponding author upon request.

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
