# Peer review of "Modification of Inconel 718 Properties by In Situ Y Addition in Selective Laser Melting"

_materials, 2022, doi:10.3390/ma15186219_

Round 1

Reviewer 1 Report

1. Why is Yttrium added?

2. What is the novelty of this work?

3. All figures - remove / hide the table gridlines

4. What is the basis for selecting 0.1%, 0.2%, 0.5%, 1% and 2% Y? Is there any scientific logic?

5. Fig 4 XRD should be indexed and all peaks to be marked with relevant crystallographic planes.

6. Fig. 5 What is the mechanism of fracture? The authors need to explain this.

7. What is the application of the current study?

8. The conclusions must be strong. The use of 'may' or 'may be' is not prudent. Authors should show scientific proof for all the points they claim in the conclusions.

9. References: A single author 'Zhou' has been cited multiple times. This has to be replaced by recent literature by other authors. 

10. typo error in heading: 3.1. Powder materials characterisctics

Author Response

Thank You for your comments and suggestions.

Responses to comments and a revised version of the manuscript in the attached documents.

Reviewer 2 Report

Dear Authors,

nice manuscript. However, I have some suggestions/criticism for changes, see comments in the attached document as well as below:

1.) the method description of particle size measurement should be detailed further, since I am not sure whether you measure the correct values for particle sizes.

2.) the conclusions with regard to mechanical properties, I do not fully support, unless you provide further references for it!

3.) Also some references worth mentioning are missing in the introduction.

4.) Also the quality of Figure 2 should be improved!

Further small issues are addressed in the attached document.

With best regards and good luck!

Author Response

(The authors gave the same response as above.)
